# Spatially congruent sites of importance for global shark and ray biodiversity

**Danielle H. Derrick** [ID]*°, **Jessica Cheok**°, **Nicholas K. Dulvy**°

Earth to Ocean Research Group, Simon Fraser University, Burnaby, British Columbia, Canada

° These authors contributed equally to this work.
* dhderric@sfu.ca

**Data Availability Statement:** All distribution files are available from the IUCN database. Adapted data from the IUCN database used for analysis have been uploaded to the primary author's GitHub repository (https://github.com/daniellederrickh)

## Abstract

Many important areas identified for conservation priorities focus on areas of high species richness, however, it is unclear whether these areas change depending on what aspect of richness is considered (e.g. evolutionary distinctiveness, endemicity, or threatened species). Furthermore, little is known of the extent of spatial congruency between biodiversity measures in the marine realm. Here, we used the distribution maps of all known marine sharks, rays, and chimaeras (class Chondrichthyes) to examine the extent of spatial congruency across the hotspots of three measures of species richness: total number of species, evolutionarily distinct species, and endemic species. We assessed the spatial congruency between hotspots considering all species, as well as on the subset of the threatened species only. We consider three definitions of hotspot (2.5%, 5%, and 10% of cells with the highest numbers of species) and three levels of spatial resolution (1°, 4°, and 8° grid cells). Overall, we found low congruency among all three measures of species richness, with the threatened species comprising a smaller subset of the overall species patterns irrespective of hotspot definition. Areas of congruency at 1° and 5% richest cells contain over half (64%) of all sharks and rays and occurred off the coasts of: (1) Northern Mexico Gulf of California, (2) USA Gulf of Mexico, (3) Ecuador, (4) Uruguay and southern Brazil, (5) South Africa, southern Mozambique, and southern Namibia, (6) Japan, Taiwan, and parts of southern China, and (7) eastern and western Australia. Coarsening resolution increases congruency two-fold for all species but remains relatively low for threatened measures, and geographic locations of congruent areas also change. Finally, for pairwise comparisons of biodiversity measures, evolutionarily distinct species richness had the highest overlap with total species richness regardless of resolution or definition of hotspot. We suggest that focusing conservation attention solely on areas of high total species richness will not necessarily contribute efforts towards species that are most at risk, nor will it protect other important dimensions of species richness.

## Introduction

Species distributions are widely used to characterise and explain the patterns seen in biodiversity throughout the world and can be used to help identify places of conservation priority [1–

under the repository title:
SpatiallyCongruentSites_2020

**Funding:** This project was funded by the National Science and Engineering Research Council of Canada (NSERC) and the Shark and Ray Conservation Fund.

**Competing interests:** The authors have declared that no competing interests exist.

3]. Species richness, defined as the number of different species in a given area, is generally greatest in the tropical latitudes [4–6]. Although this pattern is dominant in terrestrial systems, hotspots of species richness in the ocean can occur along productive frontal systems and sub-tropical boundary zones [6–8], many of which tend to result from the overlap of wider-ranging species [9]. Global assessments of biodiversity have previously focused on identifying priority areas based on total number of species alone [10], however there are other interpretations of species richness that have not yet been explored, such as evolutionary distinctiveness or endemicity.

Evolutionarily distinct (ED) species, defined as species that encompass the greatest share of evolutionary history, usually measured from the branch lengths of a phylogenetic tree [11], are also of conservation value [12]. Areas of high evolutionary distinctiveness are important to conservation because they can capture those species who embody unique forms, functions, and genomes [13]. For example, any one species of echidna embodies a greater fraction of the morphological, physiological, and ecological diversity of class Mammalia than any one species of the 2,000 or so species of rodents [12,14]. In some lineages, especially sharks and rays, extinction risk is greatest in the species that embody the largest share of this evolutionary history because they exhibit traits, such as large body size, that render them intrinsically sensitive to threats such as hunting or overfishing [13,15–17]. Endemicity is defined as those species that exist only in a defined geographic region [18]. Endemic species tend to merit high conservation priority because of their small geographical range sizes and low population numbers [19]. An influential analysis of threatened terrestrial endemics revealed that 44% of all endemic plants and 35% of endemic vertebrates occurred in only 2% of the global land area [18], demonstrating how an endemicity-centric approach can be incredibly spatially efficient in identifying areas for conservation. Identifying the geographical areas that harbor congregations for different richness metrics, such as total species, evolutionarily distinct species, or endemic species, have resulted in becoming a significant component of the terrestrial conservation agenda [18]. While there are numerous values that could be used to drive conservation, there is an urgency to conserve those threatened species that are at risk of extinction.

The 2020 Aichi biodiversity target to conserve 10% of coastal and marine areas drove a rapid expansion of marine protected areas, with the area covered rising from 0.67% of the world's oceans in 2000 to 6.4% in 2017 [20]. Within the newly drafted 2030 Kunming biodiversity framework, target 2 aims to "protect 30% of sites of particular interest on both land and sea" [21]. Now is the time to shape the rapidly developing 2030 agenda of biodiversity conservation by identifying areas that harbour the combination of the greatest richness, endemicity, and evolutionary distinctiveness [19,22,23], amongst the many other dimensions of biodiversity, as well as their threatened counterparts. In addition to shedding light on the distribution of species diversity (and across the different measures with which diversity can be defined), these identified areas can be used to inform regions of focus for subsequent systematic conservation planning exercises [24].

One quarter of all sharks, rays, and chimaeras (class Chondrichthyes; hereafter referred to as "sharks and rays") are categorized as threatened (Vulnerable, Endangered, or Critically Endangered) on the International Union for the Conservation of Nature's (IUCN) Red List of Threatened Species, or are predicted to be threatened based on their large body size and exposure to fisheries [25,26]. Sharks and rays are among the most evolutionarily distinct vertebrate radiation of marine predators [27], and their slow life histories result in low population growth rates [28–30]. These features combine to render them highly sensitive to overfishing [25,31]. The availability of comprehensive Red List Assessments and geographic distribution maps make sharks and rays a good case study to understand how marine species richness measures are spatially distributed and can be conserved most efficiently. There are few analyses that

explore the spatial distribution and overlap of different biodiversity measures in the terrestrial realm and even fewer in the ocean. The terrestrial studies have all found a lack of spatial overlap occurring throughout a variety of different taxa (i.e. birds, insects, plants) [22,32,33]. While marine studies yield comparable patterns to the terrestrial realm, most focus on relatively sessile species (i.e. coral reefs) or on other dimensions of biodiversity (i.e. functional diversity) [5,10].

Here, we use a global database of all known shark and ray distributions to explore the spatial congruency among three species richness measures: total number of species, ED species richness, and endemic species richness. Spatial congruence is defined here as the spatial overlap between hotspot areas. We also explore the level of spatial congruency of the species richness measures for threatened shark and ray species only because of their greater conservation urgency. Specifically, we examine the (1) overall spatial congruency among all species richness measures and the subset of threatened species, and (2) changes in spatial congruency according to different definitions of hotspot used, as well as different levels of spatial resolution.

## Methods

We obtained distribution maps for all known sharks, rays, and chimaeras in the class Chondrichthyes from the IUCN [25,34]. All maps were projected with Lambert equal area for analysis. A global grid map was overlain at a cell resolution of 1˚ by 1˚, equating to an approximate distance at the equator of 110 km. The global grid contains 44,181 cells after excluding terrestrial land masses, which are any cells containing land from the Environmental Systems Research Institute (ESRI) vector map of the world [35]. Across all species richness measures evaluated, each species is scored as present within a grid cell if any part of their distribution range falls within the grid cell boundaries. Total species richness ($n$ = 1,083 spp.) was calculated as the total number of unique species within each grid cell. We consider all marine species together rather than separate coastal and pelagic species because many pelagic species are also neritic–occurring on the continental shelf. Hence, we have retained the pelagic species to capture the true richness and evolutionary distinctness of shelf seas. Evolutionary distinctiveness scores were calculated as the sum of the branch lengths of a species down to the root of the phylogenetic tree, with each branch inversely weighted by the number of species that it subtends [36,37]. Species with longer branches and fewer relatives have higher evolutionary distinctiveness scores. ED species richness ($n$ = 264 spp.) was defined as those species with the highest quartile of evolutionary distinctiveness scores (represented as age in millions of years) and is calculated as the total number of unique species per cell that are within the evolutionarily distinct upper quartile. Endemic species richness ($n$ = 527 spp.) was calculated as the total number of unique species within each grid cell that have range sizes below the median of the range sizes of all species (i.e. 419,659 km$^2$) [10,38,39]. To quantify total threatened species richness ($n$ = 178 spp.), we counted the number of species within each grid cell that are currently listed as Vulnerable, Endangered, or Critically Endangered (i.e. threatened) according to the IUCN Red List Categories and Criteria [40]. Threatened endemic richness ($n$ = 70 spp.) was calculated in the same way as endemic species richness, but subset to the IUCN threatened species only. Finally, threatened ED species are those ED species that have been classified by the IUCN as threatened ($n$ = 49 spp.).

We defined richness hotspots as those containing the top 5% of richest cells for each of the biodiversity measures. Previous research has shown that the richest 1–5% of total land area can capture a substantial proportion of species [18,41,42]. We tested the extent of spatial congruency between shark and ray hotspots derived for all three species richness measures (i.e. total species, ED species, and endemic species), and between all three threatened subsets of the

biodiversity measures. Extent of spatial overlap between hotspots was calculated using the following equation [22]:

$$\text{Total proportion of overlap} = \frac{\sum C_n}{\sum A_n}$$

Where *C* is equal to the areas of congruence for each species richness measure, *A* the total distributional area of species richness measure hotspots, and *n* the number of species richness measures used to calculate congruence. To explore our original choice of hotspot (5%) or choice of spatial resolution (1˚), we also calculated spatial overlap for two different definitions of hotspot (richest 2.5% and 10% of cells), and two levels of coarser spatial resolution (4˚ and 8˚ grid cells). All analyses were carried out using ArcGIS Pro 2.4.3 [43] and R v.3.6.1 [44,45].

## Results

In general, the distributional patterns of total and ED species richness spanned the global ocean environment while endemic species were confined to the coastlines (Fig 1; S1 and S2 Figs). We focus our presentation of results and discussion of overall biodiversity patterns and congruency on the 5% definition criterion over all three resolutions (1˚, 4˚, and 8˚). The results did not greatly differ between the three definitions of species richness hotspot (Fig 2; S3–S11 Figs; S1 Table). Biodiversity hotspots for all shark and ray species were greatest near the equatorial coastlines for all measures except endemic species richness (Fig 3). There are clear deviations from the well-known latitude-richness relationship, with no species richness hotspots present around equatorial coastlines (i.e. East Africa, Central Brazil, and Central America) and some richness hotspots occurring in high latitude locations, particularly in the southern hemisphere (notably South Africa, Atlantic South America, and Australia; Fig 3A). These biodiversity patterns are more apparent for the subset of threatened species only (Fig 3D–3F). The distribution of ED species is broadly similar to the total richness pattern, but with a notable deficit along the northern coast of South America, particularly the Northwest Atlantic and eastern Pacific coastlines (Fig 3A and 3B). The anti-tropical distribution of endemicity hotspots is most strongly present in the southern hemisphere (Fig 3C and 3F).

In general, there was very low spatial congruence when comparing the hotspots of all three species richness measures (total species, ED species, endemic species; S1 Table). Cumulatively, all three biodiversity hotspots (for 1˚ resolution at 5% richest cells) occupied an area of 32,162,358 km$^2$, of which only 5.78% (1,859,971 km$^2$) were spatially congruent between all three hotspots (orange cells; Fig 4A). These eight areas of congruency occurred off the coasts of: (1) Northern Mexico Gulf of California, (2) USA Gulf of Mexico, (3) Ecuador, (4) Uruguay and southern Brazil, (5) South Africa, southern Mozambique, and southern Namibia, (6) Japan, Taiwan, and parts of southern China, and (7) eastern and western Australia (Fig 4B–4E), and in total contain over half (64%) of all marine sharks and rays. The hotspots calculated for the subset of threatened species followed a similar pattern, albeit with considerably lower spatial congruency. The hotspots derived from all biodiversity measures (at 1˚ resolution) for threatened species only covered a cumulative area of 28,839,224 km$^2$ with a mere 1.51% (436,506 km$^2$) of overlap between the three biodiversity hotspots (Fig 5A). The 1.51% of overlap occurred off the coasts of: (1) Brazil and Uruguay (making up nearly two thirds of the total area; 286,767 km$^2$), (2) South Africa, (3) Taiwan, and (4) eastern Australia (Fig 5B–5E). In total, these areas of overlap comprise 37% of all marine shark and ray species.

Of all pairwise comparisons of spatial overlap, congruency between total number of species and ED species of all shark and ray species was consistently the highest (average of ~43%), and this remained true across all definitions of hotspot, as well as levels of spatial resolution (Fig 2;

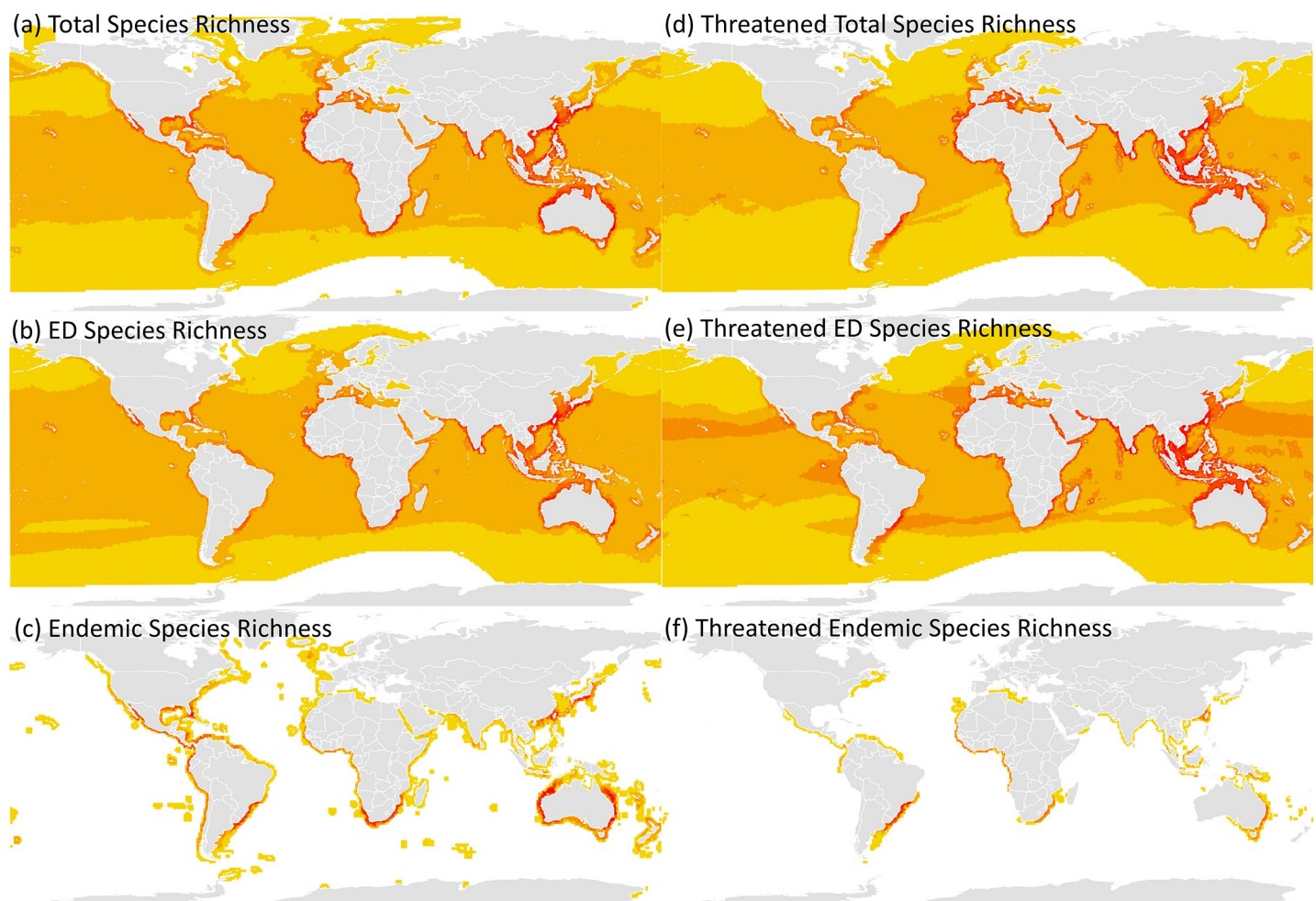

**Fig 1. Global biodiversity patterns for three measures of species richness at 1˚ resolution.** General richness for (a) total species, (b) evolutionarily distinct (ED) species, and (c) endemic species. (d-f) Threatened subsets of richness patterns for (d) total species, (e) evolutionarily distinct (ED) species, and (f) endemic species. Geographic coordinate system is in NAD83, projected coordinate system is Lambert equal area. The data used for this figure under CC BY license is granted permission from the International Union for the Conservation of Nature (IUCN), original copyright 2011.

S3 Fig; S1 Table). Conversely, spatial overlap between total number of species and endemic species of all shark and ray species remained at approximately half (average of ~20%) of the total species and ED species overlap across all definitions and resolutions of hotspot (Fig 2; S3 Fig; S1 Table). ED species and endemic species overlap followed similar low congruency trends (average of ~17%) to that of total species and endemic species (Fig 2; S3 Fig; S1 Table). The threatened species subset had similar results where ED hotspots had the highest percent of overlap with total species richness, averaging ~6% across all definitions of hotspot and levels of spatial resolution (Fig 2; S3 Fig; S1 Table). Correspondingly, spatial overlap of total species and endemic species as well as ED species and endemic species of threatened shark and ray species only, were consistently lower than congruency of total species and ED species, averaging ~4% and ~4.5% across all definitions of hotspot and levels of spatial resolutions (Fig 2; S3 Fig; S1 Table). Similar to the total species results, the highest degree of overlap for the threatened species richness subset was between total species and ED species (Fig 2; S3 Fig; S1 Table).

Our results showed that changing the definition of hotspot resulted in a minor increase in congruency between all three species richness measures, with the extent of spatial overlap still

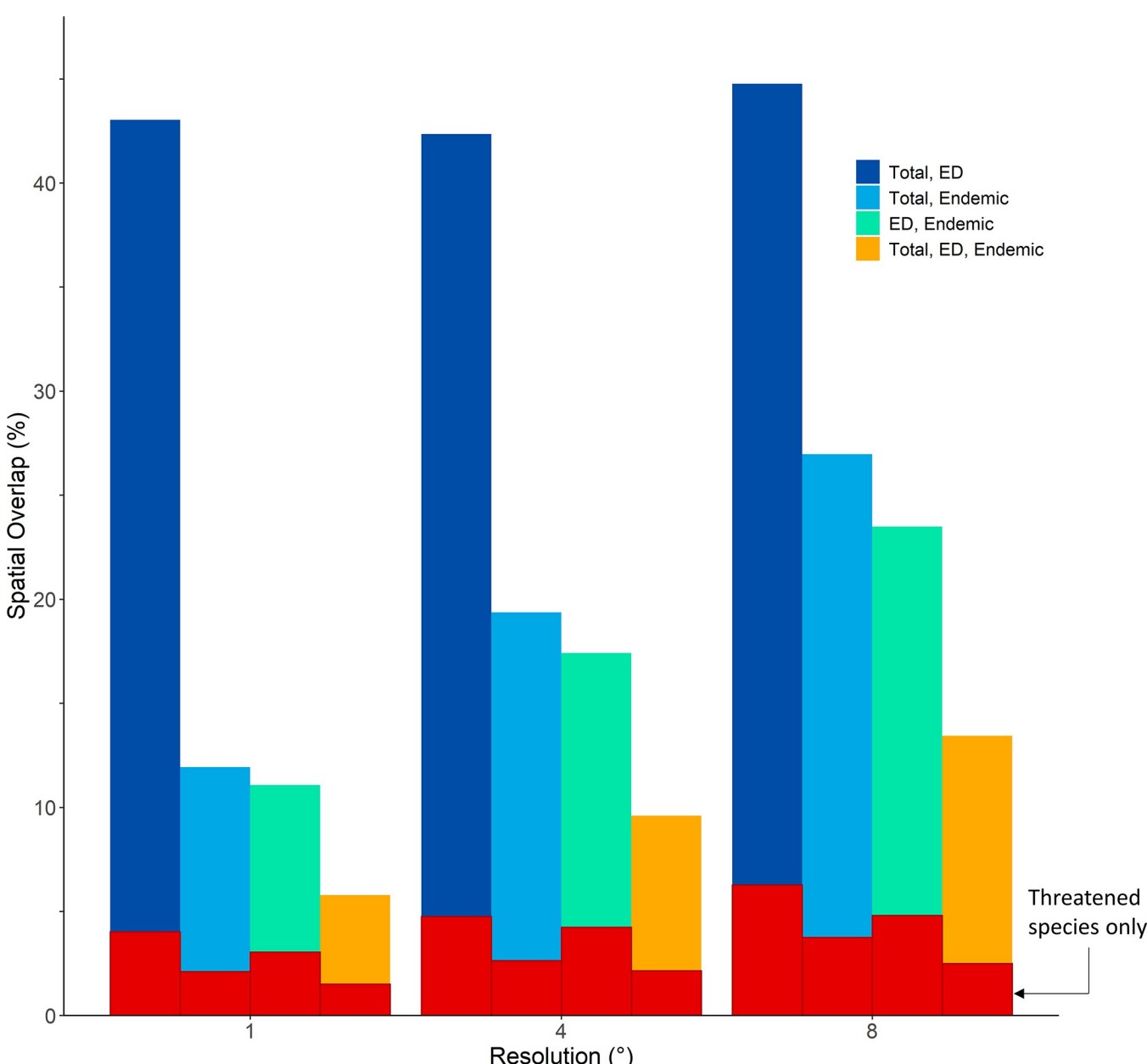

**Fig 2. Spatial congruency (measured as percent overlap) of shark hotspots between three species richness measures: Total species, evolutionary distinct (ED) species, and endemic species.** Congruency shown for hotspot definition of the richest 5% of cells and three levels of spatial resolution: 1˚, 4˚, and 8˚. The subsets of threatened species across species richness measures are indicated in red.

remaining relatively low (Fig 6A–6C; S12A–S12C Fig and S13A–S13C Fig). For example, when redefining hotspots as the richest 10% of cells, the overlap increased slightly from 5.78 to 6.38% (S1 Table). Spatial overlap for the subset of threatened species reflected similar results between hotspot definition, again displaying a minor increase when the definition of hotspot was increased (Fig 6D–6F; S12D–S12F Fig and S13D–S13F Fig). For example, at 1˚ resolution, increases in spatial overlap between the 2.5% of richest cells, 5% of richest cells, and 10% of richest cells were minor (1.04%, 1.51%, and 1.93% overlap, respectively; S1 Table).

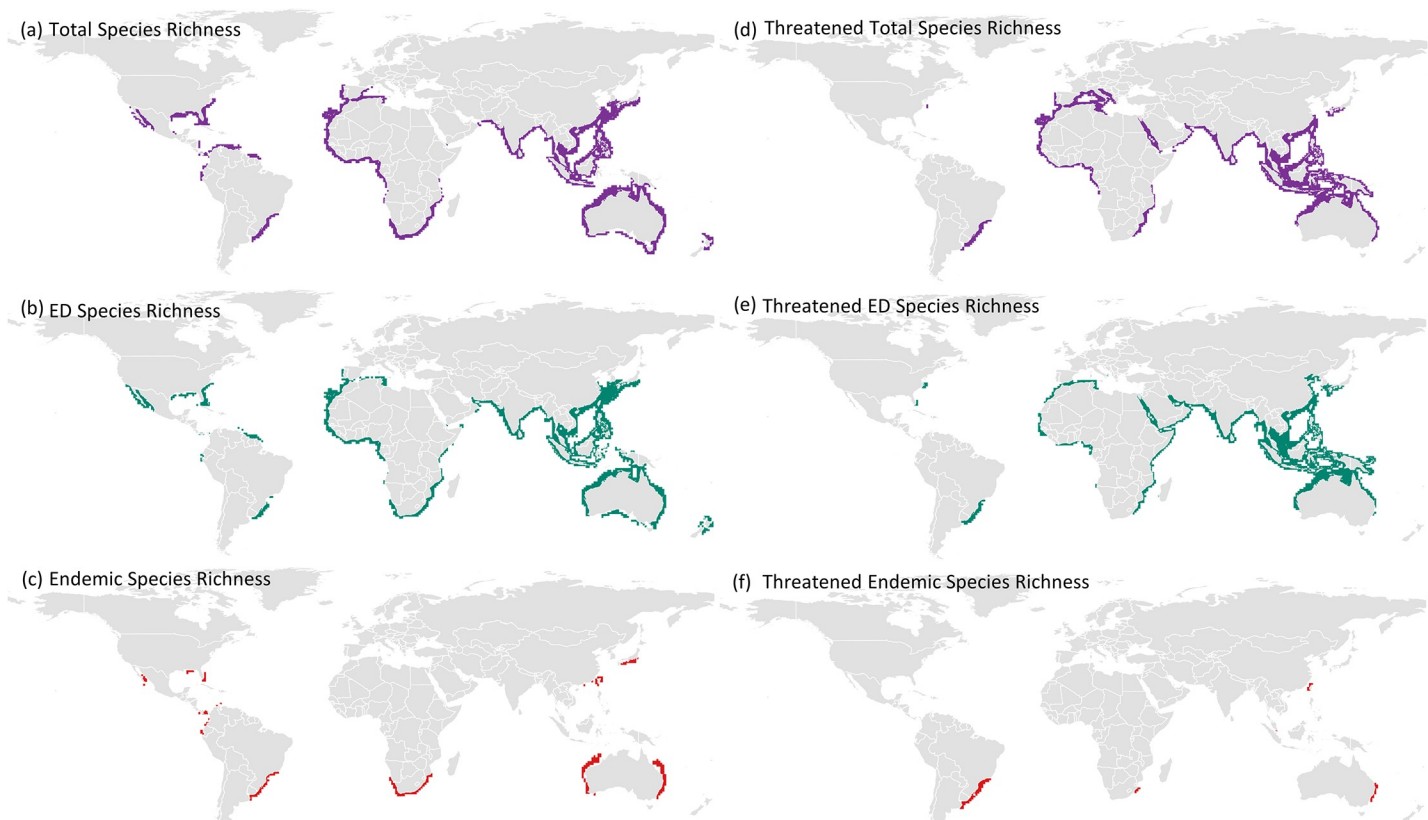

**Fig 3. Biodiversity hotspots derived for three measures of species richness.** General richness hotspots of (a) total species, (b) evolutionarily distinct (ED) species, and (c) endemic species. Richness hotspots of the threatened subset for (d) total species, (e) evolutionarily distinct (ED) species, and (f) endemic species. For each species richness measures, hotspots are defined as the richest 5% of grid cells. Geographic coordinate system is in NAD83, projected coordinate system is Lambert equal area, grid cell resolution is 1˚. The data used for this figure under CC BY license is granted permission from the International Union for the Conservation of Nature (IUCN), original copyright 2011.

Increasing the cell size from 1˚ to 8˚ led to 13.42% of hotspots being congruent, resulting in a greater than two-fold increase in congruency for all species (5.78% at 1˚ resolution), and the largest percentage of coverage contained within the country boundaries of Australia (44%), South Africa (21%), and southern Brazil and Uruguay (9.5%; Fig 6A; S1 Table). This increase in cell size also shifted the dominant locations of hotspot overlap (Fig 6A–6C). At a 4˚ resolution, areas of congruence disappeared from the coasts of Mexico and Ecuador, shifting to more representation in the USA, Colombia, and Panama (Fig 6B). At an 8˚ resolution, the spatial congruence disappeared altogether from the coasts of Brazil (Fig 6C). Similar results were seen in the threatened species subsets; despite overall low spatial overlap between levels of resolution, overlap increased marginally between 1˚, 4˚, and 8˚ cell size (1.51%, 2.15%, and 2.50% overlap, respectively; S1 Table). Spatially congruent areas between the threatened subsets were predominantly found off the coasts of southern Brazil and Uruguay (66%), which was consistent across all levels of spatial resolution examined (Fig 6D–6F). Contrastingly, these congruent areas of threatened species were present in Taiwan and Australia at 1˚ resolution, and South Africa at 1˚ and 4˚ (Fig 6D–6F). At 8˚ resolution, congruency locations for threatened species no longer corresponded at all with the areas of congruency identified for all shark and ray species (Fig 6C and 6F).

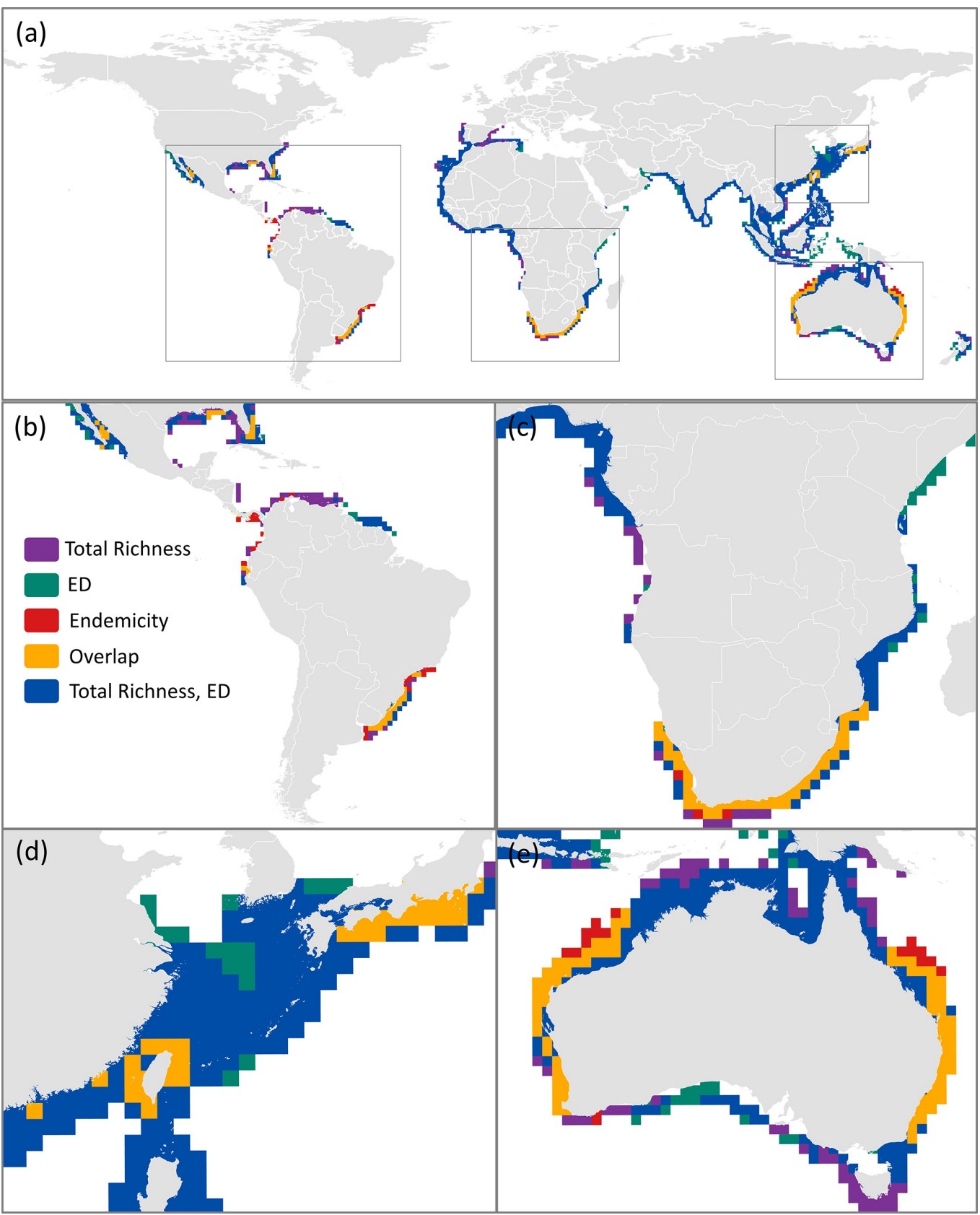

**Fig 4.** (a) Spatial congruence between global hotspots (defined at richest 5% of all grid cells) of three species richness measures: total species (purple), evolutionarily distinct (ED) species (green), and endemic species (red). Spatial congruence between hotspots derived for all three measures are represented by orange cells. Map insets highlighting specific areas of overlap: (b) North and South America, (c) southern Namibia, South Africa, and southern Mozambique, (d) Japan, Taiwan, and parts of southern China, and (e) Australia. Areas of congruence between total species richness and ED species richness are in blue. Grid cell resolution is 1˚. The data used for this figure under CC BY license is granted permission from the International Union for the Conservation of Nature (IUCN), original copyright 2011.

## Discussion

We describe four major findings. First, there was low overall spatial congruency when comparing the hotspots of all three measures of species richness (total species, ED species, and endemic species), offering a small area of focus for future conservation planning exercises. Even though those areas of spatial congruency are small in extent, they comprise approximately two thirds (64%) of all shark and ray species. Second, when comparing congruency pairwise between different species richness measures, ED species richness had the highest percent of overlap with total species richness, irrespective of spatial resolution or hotspot definition. These two findings were consistent for all shark and ray species, as well as for the subset of threatened species only. Third, congruency across the three richness measures for all threatened species is relatively insensitive to hotspot definitions (from 2.5% to 10% of richest cells) and was consistently low across these definitions. Fourth, increasing cell size (from 1˚ to 8˚) lead to a two-fold increase in congruency between all species richness measures generally. These results have implications for shark and ray biodiversity, our knowledge of the different dimensions of biodiversity and how they can differ through space, and the effect of resolution in understanding spatial congruency.

In contrast to Küper et al. [46], who demonstrated that there was a higher congruence of plant biodiversity when hotspot was redefined, we found that the extent of spatial congruency identified was low overall for the three measures of richness (total species, ED, endemic species) for all shark and ray species and the threatened species only. These results highlight considerable differences in the spatial distribution patterns of some biodiversity hotspots for sharks and rays, depending on the species richness measure used. The low congruency we have found between different measures of richness caution that it might be inappropriate to use total species richness as the sole feature of biodiversity to focus conservation attention towards. Our findings highlight that hotspots identified with other desirable species richness measures can be lost if there is a sole focus on total species richness, which has been a common strategy in identifying important areas for conservation [10,47]. If congruency among these hotspots identified with the different richness measures were high, then it would be reasonable to assume that relying on any one measure would be adequate to determine important areas for conservation that represented all three richness measures. However, our results demonstrate that this is not the case, and that not considering certain species richness measures can result in the exclusion of important features of biodiversity for conservation attention (e.g. endemic, threatened, evolutionarily distinct species). The low level of spatial congruency between the species richness measures also means that a relatively small fraction of the world's ocean area could provide a tractable focal point for global shark and ray conservation. However, we caution that this kind of focal conservation strategy would still need to account for the opportunities and challenges presented by differing social, economic and cultural contexts [48,49], in addition to the abundance, dispersal abilities, and activity patterns of the wide range of shark and ray species [50].

Interestingly, there was a relatively high spatial overlap of 43% between the hotspots identified for ED species richness and total species richness, when considering all shark and ray species. For the threatened species however, this overlap was considerably lower, at 4.02%. This

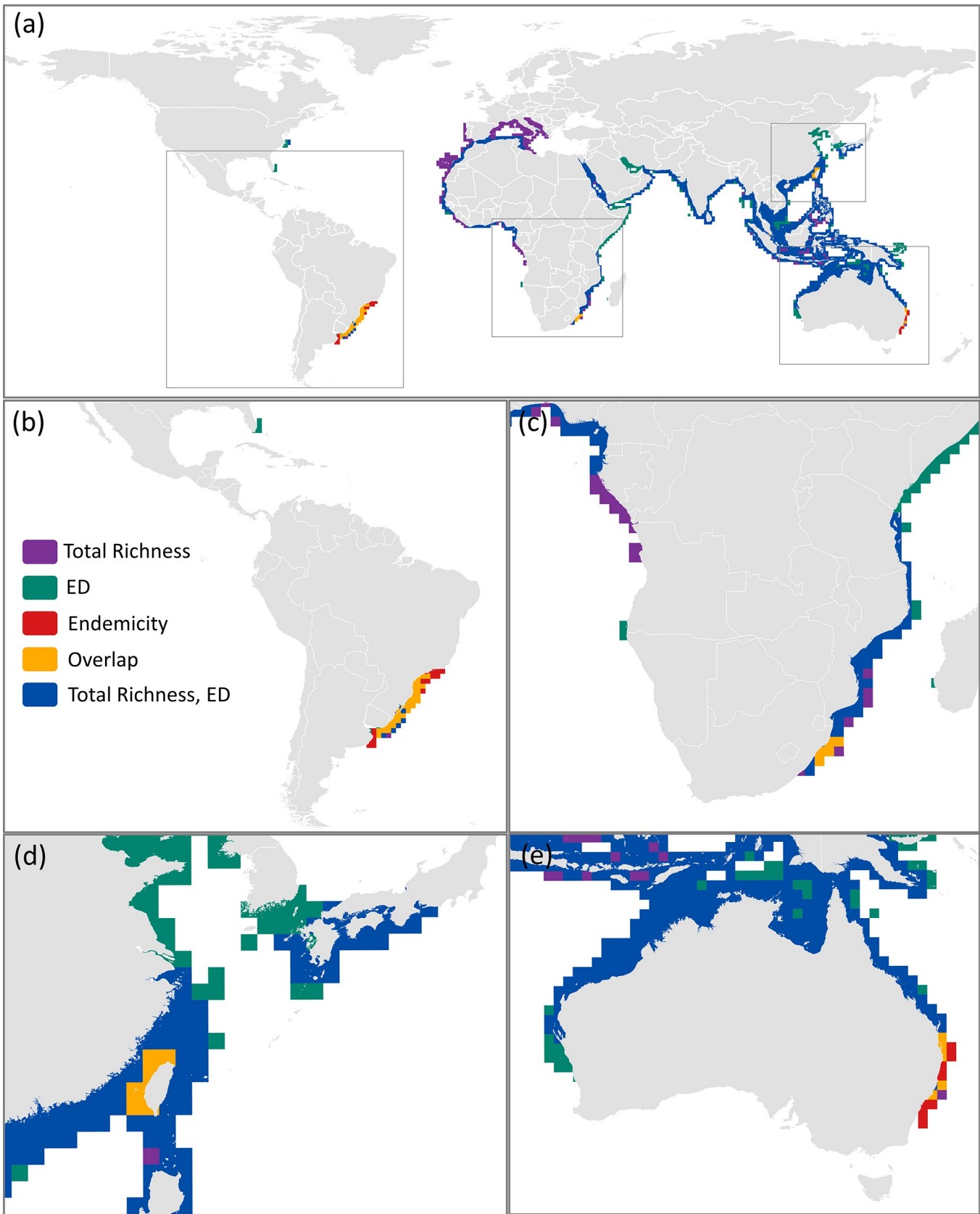

**Fig 5.** (a) Spatial congruence between threatened global hotspots (defined at richest 5% of grid cells) of three species richness measures: total species (purple), evolutionarily distinct (ED) species (green), and endemic species (red). Spatial congruence between hotspots derived for all three measures are represented by orange cells. Map insets highlighting specific areas of overlap (b) southern Brazil and Uruguay, (c) parts of South Africa, (d) Taiwan, and (e) eastern Australia. Areas of overlap between total species richness and ED species richness are in blue. Grid cell resolution is 1˚. The data used for this figure under CC BY license is granted permission from the International Union for the Conservation of Nature (IUCN), original copyright 2011.

finding of high congruency is supported by the suggestion that areas of high total species richness tend to be made-up of wide-ranging species, a characteristic commonly found in evolutionarily distinct species [9]. It is also potentially of little surprise that ED species overlap highly with total species richness because sharks and rays are one of the most evolutionarily diverse species groups with the average species embodying over 26 million years of shared unique evolutionary history [27]. Furthermore, until the last decade, it was believed that areas of high total species richness harboured both a high number of endemic and threatened species for two reasons: (1) those areas experience greater levels of threatening processes such habitat transformation and exploitation, and (2) they are likely to be inhabited by species that are on average at a greater risk to these threatening processes [1,51]. More recently however, Orme et al. [22] demonstrated weak relationships of congruence between threat and total species richness from terrestrial avian fauna, further highlighting the necessity of using different types of species richness measures to identify important areas for biodiversity conservation

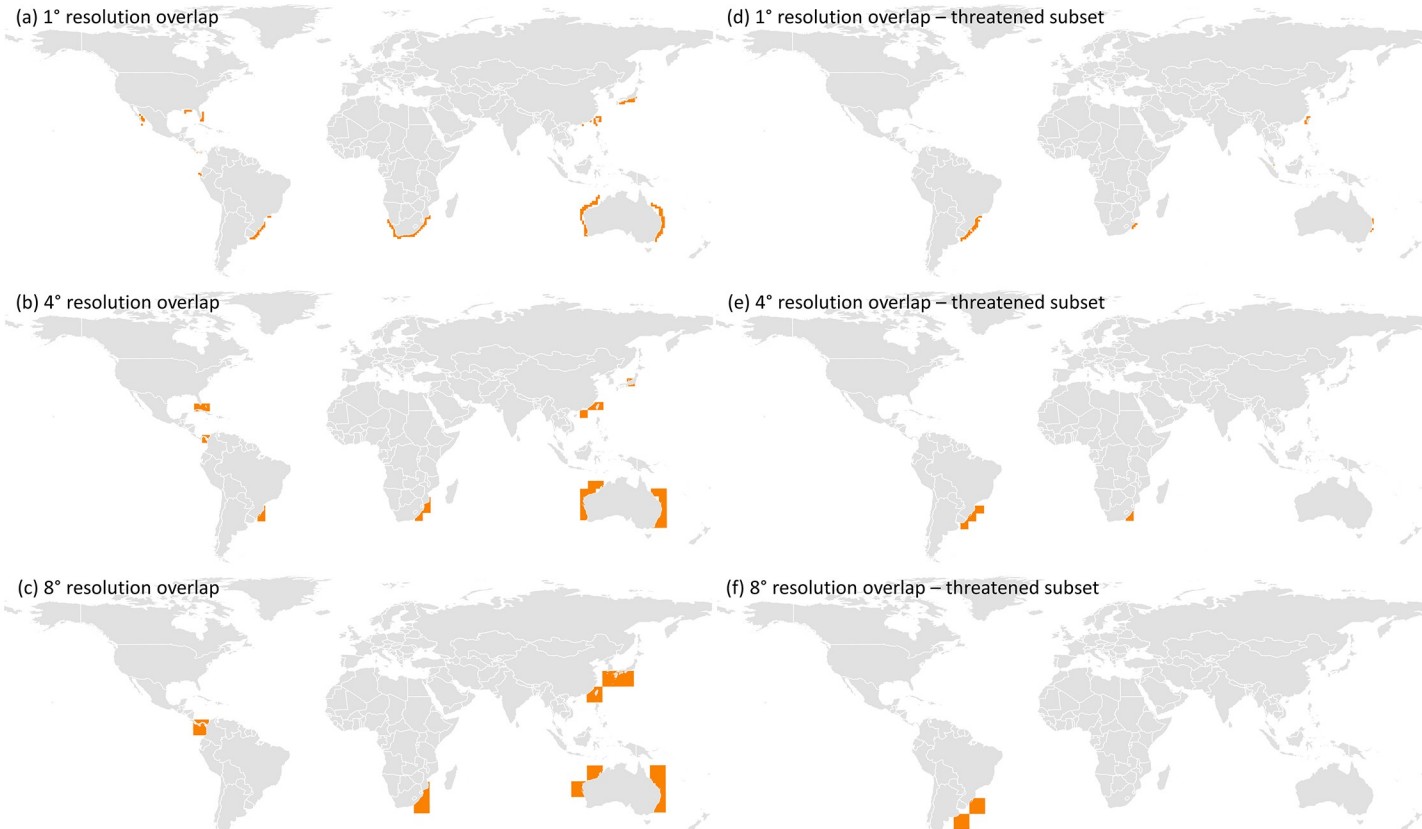

**Fig 6. Spatially congruent areas between biodiversity hotspots derived from different species richness measures represented as the richest 5% of grid all cells.** Spatially congruent areas between total species, evolutionarily distinct (ED) species, and endemic species at resolution levels of (a) 1˚, (b) 4˚, and (c) 8˚, and (d-f) congruent areas for the threatened species subsets at each corresponding resolution level. The data used for this figure under CC BY license is granted permission from the International Union for the Conservation of Nature (IUCN), original copyright 2011.

[22]. Our study is one of the first to demonstrate a relatively high degree of spatial congruence between hotspots of ED species richness and total species richness of all shark and ray species, as compared to the overlap of endemic species and total species.

The areas of spatial congruence for total and threatened shark and ray species cluster around coastal waters, while endemic species are primarily found at the convergent boundaries of tropical and temperate ecosystems. These warm reef environments at the convergent boundaries have been known to serve as hotspots for species evolution due to their high productivity and habitat complexity [52,53]. In most cases, these areas of overlap are also found within the bounds of a country's exclusive economic zone (EEZ), which have also been flagged as hotspots of functional diversity in sharks [5]. The species richness measures examined in this paper only represent a small aspect of biodiversity and do not take into account other measures, such as functional diversity. Functional diversity is known to be crucial in maintaining the structure and function of marine ecosystems [54] and would likely also yield similarly incongruent hotspots. Ultimately, a future study could expand on our findings by exploring the extent of spatial congruency between other biodiversity metrics, such as functional diversity in all sharks and rays.

Studies that consider different levels of spatial resolutions have considered only one level of resolution that are either smaller (e.g. $\leq 1°$) [5,23,55] or larger (e.g. $\geq 8°$) [4] than those assessed in our study, missing the potential differences that could occur between the two. Our findings demonstrate that there are differences between these two levels of spatial resolution. We found that a reduction in resolution (i.e. larger sampling units, such as grid cells here) influenced global patterns of species richness hotspots for all sharks and rays. For example, at a coarse resolution (here, 8° cells), if an individual species' range slightly crossed the boundary of an 8° grid cell, its distribution would now be considered to encompass the entirety of that 8° cell as opposed to its true smaller fraction. The coarsening of hotspots and shifting of congruency locations resulting from coarser resolutions causes congruency locations to disappear where they were otherwise present at finer resolutions (i.e. Brazil and Uruguay; Fig 6). Previous work on riparian weeds also found that coarser resolutions were unable to model fine-scale distributions successfully and were also poor predictors of national species' distributions [56]. Overall, our results support the well-known finding that changes in spatial resolution can influence results in spatial analyses. Different areas of congruency identified at various spatial resolutions can make it difficult for conservation management to direct focus to any particular area but demonstrates the importance of explicitly considering spatial resolution when determining important areas to further investigate for conservation priority. Furthermore, there are now numerous studies that examine how to integrate conservation planning across multiple levels of resolution [57–59].

It is important to note the caveats of the distributional dataset used for this study. The IUCN species distribution map database was created from peer-reviewed, expert-generated maps around known locations of species distributions [25]. Experts from the IUCN Shark Specialist Group (SSG) created a shapefile of the geographic distribution for each chondrichthyan species based on the original maps provided to the Food and Agriculture Organisation of the United Nations, using the standard mapping protocol for marine species devised by the IUCN Global Marne Species Assessment team (https://sites.wp.odu.edu/GMSA/). The maps show the Extent of Occurrence of the species cut to one of several standardized basemaps depending on the ecology of the species (i.e. coastal and continental shelf, pelagic, and deepwater). The original maps were updated, corrected, or verified by experts at the Red List workshops or by out-of-session assessors and SSG staff [25]. These maps are likely to contain commission rather than omission errors such that a species is shown to be present in an area when in fact it is not [60]. Commission errors can be problematic for hotspot identification because they risk

identifying areas that are not true hotspots and directing valuable and limited conservation resources to those untrue hotspot areas [61]. Omission errors risk missing true hotspot areas of richness and therefore true areas of congruency between the different species richness metrics. Omission errors can also result in a reduction of spatial options available when it comes to systematic conservation planning [62]. Aqua-maps can be used as an alternative or complementary data source to the IUCN distribution maps, they are created using habitat suitability models based on point distribution data and thus give an indication of probabilities of species occurrence across the distribution ranges [63]. However, these models are rarely vetted by taxonomists that understand the biology and geographical distribution and veracity of point records. Although the IUCN distributional data are not without limitations, they are currently the most comprehensive datasets for studying shark and ray biodiversity patterns in the ocean. While we recognize there have been range contractions, our approach is to identify the historic pattern of richness for each species and demonstrate a baseline understanding of global shark and ray biodiversity [64,65]. These maps are continually refined with routine updates of global species catalogues and field guides, lending scope to conduct more refined global analyses in future studies [66–69].

Although this was in essence a global analysis, the low richness and wide ranging nature of species inevitably means no hotspots were found in the pelagic ecosystem. Furthermore, endemic species richness tends to be strictly coastal, unless defined differently than the one used in this study. Therefore, future work can examine the identification of hotspot areas of species richness measures and their corresponding areas of spatial congruency when coastal and pelagic ecosystems are analyzed independently. A lack of spatial congruency among the three species richness measures also opens up future work to explore the potential differences in environmental and evolutionary drivers of individual species richness measures, at varying spatial extents. For example, at smaller extents (e.g. local) species have been known to be influenced by local attributes like competition, and habitat availability, whereas at large extents (e.g. global) it is hypothesized that environmental variables have a stronger relationship with global species patterns [8,70,71]. In conclusion, the lack of spatial congruency between different species richness measures (and likely other biodiversity measures) could provide a global informative perspective on areas that merit further attention where management could focus their efforts for the conservation of shark and ray biodiversity, especially in preparation for the 2030 Kunming Targets. The low level of spatial congruency means that the eight places with spatial overlap in all three measures of species richness might provide a useful starting point to direct conservation planning, Marine Protected Area designation, and improved fisheries management and secure a future for sharks and rays.

## Supporting information

**S1 Fig. Global biodiversity patterns for three measures of species richness at 4˚ resolution.** General richness for (a) total species, (b) evolutionarily distinct (ED) species, and (c) endemic species. (d-f) Threatened subsets of richness patterns for (d) total species, (e) evolutionarily distinct (ED) species, and (f) endemic species. Geographic coordinate system is in NAD83, projected coordinate system is lambert equal area. The data used for this figure under CC BY license is granted permission from the International Union for the Conservation of Nature (IUCN), original copyright 2011.
(DOCX)

**S2 Fig. Global biodiversity patterns for three measures of species richness at 8˚ resolution.** General richness for (a) total species, (b) evolutionarily distinct (ED) species, and (c) endemic species. (d-f) Threatened subsets of richness patterns for (d) total species, (e) evolutionarily

distinct (ED) species, and (f) endemic species. Geographic coordinate system is in NAD83, projected coordinate system is lambert equal area. The data used for this figure under CC BY license is granted permission from the International Union for the Conservation of Nature (IUCN), original copyright 2011.
(DOCX)

**S3 Fig. Spatial congruency (measured as percent overlap) of shark hotspots between three species richness measures: Total species, evolutionary distinct (ED) species, and endemic species.** Congruency is represented at two levels of hotspot definition: (a) 2.5% and (b) 10%, and three levels of spatial resolution: 1˚, 4˚, and 8˚. The subset of threatened species only are indicated in red.
(DOCX)

**S4 Fig. Biodiversity hotspots derived for three measures of species richness.** General richness hotspots of (a) total species, (b) evolutionarily distinct (ED) species, and (c) endemic species. (d-f) Threatened subset of richness hotspots for (d) total species, (e) evolutionarily distinct (ED) species, and (f) endemic species. For each species richness measures, hotspots are defined as the richest 2.5% of grid all cells. Geographic coordinate system is in NAD83, projected coordinate system is lambert equal area, grid cell resolution is 1˚. The data used for this figure under CC BY license is granted permission from the International Union for the Conservation of Nature (IUCN), original copyright 2011.
(DOCX)

**S5 Fig. Biodiversity hotspots derived for three measures of species richness.** General richness hotspots of (a) total species, (b) evolutionarily distinct (ED) species, and (c) endemic species. (d-f) Threatened subset of richness hotspots for (d) total species, (e) evolutionarily distinct (ED) species, and (f) endemic species. For each species richness measures, hotspots are defined as the richest 10% of grid cells. Geographic coordinate system is in NAD83, projected coordinate system is lambert equal area, grid cell resolution is 1˚. The data used for this figure under CC BY license is granted permission from the International Union for the Conservation of Nature (IUCN), original copyright 2011.
(DOCX)

**S6 Fig. Biodiversity hotspots derived for three measures of species richness.** General richness hotspots of (a) total species, (b) evolutionarily distinct (ED) species, and (c) endemic species. (d-f) Threatened subset of richness hotspots for (d) total species, (e) evolutionarily distinct (ED) species, and (f) endemic species. For each species richness measures, hotspots are defined as the richest 2.5% of grid cells. Geographic coordinate system is in NAD83, projected coordinate system is lambert equal area, grid cell resolution is 4˚. The data used for this figure under CC BY license is granted permission from the International Union for the Conservation of Nature (IUCN), original copyright 2011.
(DOCX)

**S7 Fig. Biodiversity hotspots derived for three measures of species richness.** General richness hotspots of (a) total species, (b) evolutionarily distinct (ED) species, and (c) endemic species. (d-f) Threatened subset of richness hotspots for (d) total species, (e) evolutionarily distinct (ED) species, and (f) endemic species. For each species richness measures, hotspots are defined as the richest 5% of grid cells. Geographic coordinate system is in NAD83, projected coordinate system is lambert equal area, grid cell resolution is 4˚. The data used for this figure under CC BY license is granted permission from the International Union for the Conservation

of Nature (IUCN), original copyright 2011.
(DOCX)

**S8 Fig. Biodiversity hotspots derived for three measures of species richness.** General richness hotspots of (a) total species, (b) evolutionarily distinct (ED) species, and (c) endemic species. (d-f) Threatened subset of richness hotspots for (d) total species, (e) evolutionarily distinct (ED) species, and (f) endemic species. For each species richness measures, hotspots are defined as the richest 10% of grid cells. Geographic coordinate system is in NAD83, projected coordinate system is lambert equal area, grid cell resolution is 4˚. The data used for this figure under CC BY license is granted permission from the International Union for the Conservation of Nature (IUCN), original copyright 2011.
(DOCX)

**S9 Fig. Biodiversity hotspots derived for three measures of species richness.** General richness hotspots of (a) total species, (b) evolutionarily distinct (ED) species, and (c) endemic species. (d-f) Threatened subset of richness hotspots for (d) total species, (e) evolutionarily distinct (ED) species, and (f) endemic species. For each species richness measures, hotspots are defined as the richest 2.5% of grid cells. Geographic coordinate system is in NAD83, projected coordinate system is lambert equal area, grid cell resolution is 8˚. The data used for this figure under CC BY license is granted permission from the International Union for the Conservation of Nature (IUCN), original copyright 2011.
(DOCX)

**S10 Fig. Biodiversity hotspots derived for three measures of species richness.** General richness hotspots of (a) total species, (b) evolutionarily distinct (ED) species, and (c) endemic species. (d-f) Threatened subset of richness hotspots for (d) total species, (e) evolutionarily distinct (ED) species, and (f) endemic species. For each species richness measures, hotspots are defined as the richest 5% of grid cells. Geographic coordinate system is in NAD83, projected coordinate system is lambert equal area, grid cell resolution is 8˚. The data used for this figure under CC BY license is granted permission from the International Union for the Conservation of Nature (IUCN), original copyright 2011.
(DOCX)

**S11 Fig. Biodiversity hotspots derived for three measures of species richness.** General richness hotspots of (a) total species, (b) evolutionarily distinct (ED) species, and (c) endemic species. (d-f) Threatened subset of richness hotspots for (d) total species, (e) evolutionarily distinct (ED) species, and (f) endemic species. For each species richness measures, hotspots are defined as the richest 10% of grid cells. Geographic coordinate system is in NAD83, projected coordinate system is lambert equal area, grid cell resolution is 8˚. The data used for this figure under CC BY license is granted permission from the International Union for the Conservation of Nature (IUCN), original copyright 2011.
(DOCX)

**S12 Fig. Spatially congruent areas between biodiversity hotspots derived from different species richness measures represented as the richest 2.5% of grid cells.** Spatially congruent areas between total species, ED species, and endemic species at (a) 1˚ resolution, (b) 4˚ resolution, and (c) 8˚ resolution, and (d-f) for the subset of threatened species, corresponding to resolution levels of (a-c). The data used for this figure under CC BY license is granted permission from the International Union for the Conservation of Nature (IUCN), original copyright 2011.
(DOCX)

**S13 Fig. Spatially congruent areas between biodiversity hotspots derived from different species richness measures represented as the richest 10% of grid cells.** Spatially congruent areas between total species, ED species, and endemic species at (a) 1° resolution, (b) 4° resolution, and (c) 8° resolution, and (d-f) for the subset of threatened species, corresponding to resolution levels of (a-c). The data used for this figure under CC BY license is granted permission from the International Union for the Conservation of Nature (IUCN), original copyright 2011. (DOCX)

**S1 Table. Spatial congruency (measured as percent overlap) of shark and ray hotspots between three species richness measures: Total species, evolutionary distinct (ED) species, and endemic species.** Congruency is compared between three levels of spatial resolution: 1°, 4°, and 8°, for total number of species and the subset of threatened species, and at three levels of defining hotspot (2.5%, 5%, and 10% of richest cells). (DOCX)

## Acknowledgments

We thank all of the IUCN SSG members and all additional experts who have contributed their data and expertise to the IUCN Red List assessments of all Chondrichthyans. We also thank all members of the Dulvy Lab for helpful input.

## Author Contributions

**Conceptualization:** Danielle H. Derrick.

**Data curation:** Danielle H. Derrick.

**Formal analysis:** Danielle H. Derrick.

**Funding acquisition:** Nicholas K. Dulvy.

**Methodology:** Danielle H. Derrick.

**Supervision:** Nicholas K. Dulvy.

**Validation:** Jessica Cheok.

**Writing – original draft:** Danielle H. Derrick.

**Writing – review & editing:** Danielle H. Derrick, Jessica Cheok, Nicholas K. Dulvy.

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
