## [Decision Letter · Decision Letter 0]

22 Apr 2020

PONE-D-20-05961

Spatially congruent sites of importance for global shark and ray biodiversity

PLOS ONE

Dear Miss Derrick,

Thank you for submitting your manuscript to PLOS ONE. After careful consideration, we feel that it has merit but does not fully meet PLOS ONE’s publication criteria as it currently stands. Therefore, we invite you to submit a revised version of the manuscript that addresses the points raised during the review process.

Please address all comments by both reviewers and me when revising your manuscript.

We would appreciate receiving your revised manuscript by Jun 06 2020 11:59PM. To enhance the reproducibility of your results, we recommend that if applicable you deposit your laboratory protocols in protocols.io, where a protocol can be assigned its own identifier (DOI) such that it can be cited independently in the future. For instructions see: http://journals.plos.org/plosone/s/submission-guidelines#loc-laboratory-protocols

We look forward to receiving your revised manuscript.

Kind regards,

William David Halliday, Ph.D.

Academic Editor

PLOS ONE

Additional Editor Comments (if provided):

This paper is generally well written, and both reviewers and I agree that it presents an interesting analysis. Both reviewers brought up important points that must be addressed. In addition to these, I found a few minor issues:

Lines 33-34: Most of these locations aren't particularly useful, especially for countries that have coastlines in more than one ocean. Please be more specific.

Line 53: This is circular. You say that "species-rich" areas are commonly identified, but then suggest that other dimensions of the exact same thing might be required. How about "other indices of conservation values" or "other metrics of biodiversity"?

Line 116: "were obtained" should be deleted.

Lines 249-250: Used "resulted" and "resulting" in the same sentence. Find a new word to replace one of the "result-" words.

2. We note that Figures 2-5 and S1-S10 in your submission contain map images which may be copyrighted. All PLOS content is published under the Creative Commons Attribution License (CC BY 4.0), which means that the manuscript, images, and Supporting Information files will be freely available online, and any third party is permitted to access, download, copy, distribute, and use these materials in any way, even commercially, with proper attribution. For these reasons, we cannot publish previously copyrighted maps or satellite images created using proprietary data, such as Google software (Google Maps, Street View, and Earth). For more information, see our copyright guidelines: http://journals.plos.org/plosone/s/licenses-and-copyright.

1.    You may seek permission from the original copyright holder of Figures 2-5 and S1-S10 to publish the content specifically under the CC BY 4.0 license. 

3. Please include a copy of Table 1 which you refer to in your text on page 9 and 11-12.

Reviewers' comments:

Reviewer's Responses to Questions

**Comments to the Author**

1. Is the manuscript technically sound, and do the data support the conclusions?

Reviewer #1: Yes

Reviewer #2: Yes

2. Has the statistical analysis been performed appropriately and rigorously? 

Reviewer #1: Yes

Reviewer #2: N/A

3. Have the authors made all data underlying the findings in their manuscript fully available?

Reviewer #1: No

Reviewer #2: Yes

4. Is the manuscript presented in an intelligible fashion and written in standard English?

Reviewer #1: Yes

Reviewer #2: Yes

5. Review Comments to the Author

Reviewer #1: A review of “Spatially congruent sites of importance for global shark and ray biodiversity”

This manuscript describes important research pertaining to the spatial congruency of marine hotspots for Chondrichthyan species and has timely applications to the development of conservation plans for the 2030 Kunming Targets. In addition to being a valuable contribution to the scientific community, the manuscript is clear, concisely written, and I believe it will be accessible to all PLOS One readers. As such I recommend that this this manuscript should be accepted for publication pending minor revisions, which I have listed here:

- As per the PLOS One requirements I need to make a note that the data used in this manuscript have not been provided to the reader in the manuscript, supplementary materials, or in a repository (though the authors did explain where the data came from and noted that they will be provided once the manuscript is accepted)

- The authors use the terms “congruence” and “congruency” throughout the manuscript, but they are not clearly defined. I think it would improve the readability of the manuscript if the authors would clearly define these terms (e.g. Line 142)

- Line 67: there may be an extra space between the words “is defined”

- Lines 69-73: this sentence is a bit of a run-on and is confusing to read so I suggest the authors revise it.

- Lines 73-74: “species congregations of total species richness” is also confusing to read, I’d suggest the authors rephrase this section of the sentence.

- Line 166: remove words “were obtained”

- Figure 1: I don’t believe showing all 9 plots is necessary in the main manuscript. I’d suggest only keeping the plots that are essential to understanding the results (i.e. the 5% plots for each resolution) and to move the remainder of the plots to the supplementary section. Additionally, I think the addition of minor ticks on the y axis would improve readability of these plots.

- Line 158: please clarify what resolution you are referring to

- Line 169: typo? I think the authors meant 2F

- Figure 2: Unless this is opposed by PLOS One’s formatting requirements, I think the addition of subheadings describing each map would help to clarify this figure.

- Line 185: Where is Table 1? I suspect this is a typo and it should say S1? If this is not a typo Table 1 was missing from the reviewer materials. Table 1 is also mentioned on lines 236, 241, and 252.

- Line 195: space missing between “The” and “1.51%”

- Figure 3 & 4 (and corresponding supplementary figures): I think adding a colour legend to the maps would help to clarify the figures (instead of identifying different colours in individual maps)

- Figure 5 (and corresponding supplementary figures): The yellow colour chosen here is hard to see, especially when only a few cells are coloured (such as in East Asia). If the authors could change the colour to something more readable (perhaps a darker orange) it would greatly improve the figures. I think subheading would greatly help here as well (if possible).

Reviewer #2: Overview:

This paper presents a well-written and very straightforward overlap analysis between different diversity measures for elasmobranchs, i.e. shark, rays and chimeras. It addresses a question that has been asked and answered for other species groups but not yet for elasmobranchs: whether hotspots of species richness also capture evolutionarily distinct, endemic or threatened species reasonably well. The authors find that evolutionary distinctiveness is well captured by total richness, but endemicity less so, and threatened species even less so. This adds to an ongoing discussion about using species richness as a proxy for conservation planning. The authors also identify a few areas where all measures overlap well; although these areas are small in extent they cover almost two thirds (64%) of all elasmobranch species. I think more could be made of this latter finding, which actually surprised me. These results is highly relevant for global conservation planning.

Major concerns:

My two major concerns are as follows:

1. The analysis is entirely focused on coastal species, because most of elasmobranch diversity is coastal. Earlier work has shown, however, that coastal and pelagic species show very different diversity patterns (e.g. Tittensor et al. 2010, Nature). They also likely face different threats and are managed by different entities (i.e. national governments vs. RFMOs). Therefore I think that the analysis should be repeated by separating coastal from pelagic species. The latter will show quite different patterns of richness and different overlap patterns.

2. The analysis is entirely based on IUCN range data which are (as far as I know) largely based on expert opinion. The discussion needs a thorough section on caveats and limitations in this data source. As it, it is treated without any reservations. But it’s unlikely that these maps represent current richness, especially for threatened species which are likely to have undergone significant range extensions. I urge the author to think about how these limitations may have impacted their analysis.

Minor comments:

L41 .. other dimensions of interpretations (not sue what is meant here, revise wording)

L80/81; Please update those figures to 2020 using WCMC data base, MPA Atlas or both

Introduction is well written and comprehensive, could be tightened up a bit

Figure 1 is hard to read and understand, please think about ways to improve clarity.

6. PLOS authors have the option to publish the peer review history of their article (what does this mean?). If published, this will include your full peer review and any attached files.

Reviewer #1: No

Reviewer #2: No

---

## [Author Response · Author response to Decision Letter 0]

2 Jun 2020

Response to the editor and reviewer are provided in the 'Response to Reviewers' document that was uploaded and are also outlined below:

Additional Editor Comments (if provided):

This paper is generally well written, and both reviewers and I agree that it presents an interesting analysis. Both reviewers brought up important points that must be addressed. In addition to these, I found a few minor issues:

Lines 33-34: Most of these locations aren't particularly useful, especially for countries that have coastlines in more than one ocean. Please be more specific.

Agreed, changed to: “Areas of congruency at 1° and 5% richest cells contain over half (64%) of all sharks and rays and occurred off the coasts of: (1) Northern Mexico Gulf of California, (2) USA Gulf of Mexico, (3) Ecuador, (4) Uruguay and southern Brazil, (5) South Africa, southern Mozambique, and southern Namibia, (6) Japan, Taiwan, and parts of southern China, and (7) eastern and western Australia.”

Line 53: This is circular. You say that "species-rich" areas are commonly identified, but then suggest that other dimensions of the exact same thing might be required. How about "other indices of conservation values" or "other metrics of biodiversity"?

The idea of this sentence is to highlight that there are different ways of interpreting species richness, but previous work tends to solely focus on total number of species and not other metrics. Sentence changed to: “Global assessments of biodiversity have previously focused on identifying priority areas based on total number of species alone [10], however there are other interpretations of species richness that have not yet been explored, such as evolutionary distinctiveness or endemicity.”

Line 116: "were obtained" should be deleted.

Done.

Lines 249-250: Used "resulted" and "resulting" in the same sentence. Find a new word to replace one of the "result-" words.

Changed to “Increasing the cell size from 1° to 8° led to 13.42% of hotspots being congruent, resulting in a near two-fold…”

Manuscript was checked and changes were made where required (i.e. change on line 242 to Fig 5D-F, changing Table 1 to S1 Table, changes made to fix some of the references).

2. We note that Figures 2-5 and S1-S10 in your submission contain map images which may be copyrighted. All PLOS content is published under the Creative Commons Attribution License (CC BY 4.0), which means that the manuscript, images, and Supporting Information files will be freely available online, and any third party is permitted to access, download, copy, distribute, and use these materials in any way, even commercially, with proper attribution. For these reasons, we cannot publish previously copyrighted maps or satellite images created using proprietary data, such as Google software (Google Maps, Street View, and Earth). For more information, see our copyright guidelines: http://journals.plos.org/plosone/s/licenses-and-copyright.

The maps are not copyrighted. Vector files that represent land masses were taken from Natural Earth data (cited as reference 37 in the methods section), and species distribution shapefiles used to create hotspots were taken from the IUCN database (cited as reference 25 and 34 in the methods section). All figures were produced by the authors specifically for this manuscript.

1. You may seek permission from the original copyright holder of Figures 2-5 and S1-S10 to publish the content specifically under the CC BY 4.0 license. 

3. Please include a copy of Table 1 which you refer to in your text on page 9 and 11-12.

We apologize, Table 1 in the manuscript should be referenced as S1 Table, as part of the Supplementary Materials. Appropriate changes have been made to the manuscript. 

Reviewers' comments:

Reviewer's Responses to Questions

Comments to the Author

1. Is the manuscript technically sound, and do the data support the conclusions?

Reviewer #1: Yes

Reviewer #2: Yes

2. Has the statistical analysis been performed appropriately and rigorously? 

Reviewer #1: Yes

Reviewer #2: N/A

3. Have the authors made all data underlying the findings in their manuscript fully available?

Reviewer #1: No

Reviewer #2: Yes

4. Is the manuscript presented in an intelligible fashion and written in standard English?

Reviewer #1: Yes

Reviewer #2: Yes 

5. Review Comments to the Author

Reviewer #1: A review of “Spatially congruent sites of importance for global shark and ray biodiversity”

This manuscript describes important research pertaining to the spatial congruency of marine hotspots for Chondrichthyan species and has timely applications to the development of conservation plans for the 2030 Kunming Targets. In addition to being a valuable contribution to the scientific community, the manuscript is clear, concisely written, and I believe it will be accessible to all PLOS One readers. As such I recommend that this this manuscript should be accepted for publication pending minor revisions, which I have listed here:

- As per the PLOS One requirements I need to make a note that the data used in this manuscript have not been provided to the reader in the manuscript, supplementary materials, or in a repository (though the authors did explain where the data came from and noted that they will be provided once the manuscript is accepted)

The only data used for this analysis were the distribution maps provided by the IUCN. These maps are published on an open access website and freely available for non-commercial use (https://www.iucnredlist.org/terms/terms-of-use), which we have cited appropriately in our manuscript. Therefore, these maps do not need to be provided in a repository. 

- The authors use the terms “congruence” and “congruency” throughout the manuscript, but they are not clearly defined. I think it would improve the readability of the manuscript if the authors would clearly define these terms (e.g. Line 142)

We have added a definition, to the first introduction of the concept in the introduction (lines 117 – 118 in the marked up document): “Spatial congruence is defined here as the spatial overlap between hotspot areas.”

- Line 67: there may be an extra space between the words “is defined”

This was checked, there is no extra space between the words. However, to be sure, we also checked the whole manuscript for double spaces and corrected all that were found. 

- Lines 69-73: this sentence is a bit of a run-on and is confusing to read so I suggest the authors revise it.

Changed to: “An influential analysis of threatened terrestrial endemics revealed that 44% of all endemic plants and 35% of endemic vertebrates occurred in only 2% of the global land area [18], demonstrating how an endemic-centric approach can be incredibly spatially efficient in identifying areas for conservation.” We have added punctuation to break up the sentence and reworded for conciseness.

- Lines 73-74: “species congregations of total species richness” is also confusing to read, I’d suggest the authors rephrase this section of the sentence.

Changed to: “Identifying the geographical areas that harbor species congregations for metrics such as total species richness, evolutionarily distinct species richness, or endemic species richness have resulted in becoming a significant component of the terrestrial conservation agenda [18,20].”

- Line 166: remove words “were obtained”

There was no “were obtained” words at this line reference. However, if they are referring to line 116, that has been updated and removed. 

- Figure 1: I don’t believe showing all 9 plots is necessary in the main manuscript. I’d suggest only keeping the plots that are essential to understanding the results (i.e. the 5% plots for each resolution) and to move the remainder of the plots to the supplementary section. Additionally, I think the addition of minor ticks on the y axis would improve readability of these plots.

Figure 1 (now Figure 2) has been updated to include only the 5% plots for all three resolution levels while the other two panels have been moved to the SOM as S3 Fig. Minor tick marks on the y-axis were also added.

- Line 158: please clarify what resolution you are referring to

This has been updated.

- Line 169: typo? I think the authors meant 2F

Yes, thank you. It was a typo, corrected to 2F.

- Figure 2: Unless this is opposed by PLOS One’s formatting requirements, I think the addition of subheadings describing each map would help to clarify this figure.

Descriptive subheadings have been added to Figure 2 (now Figure 3), as well as other similar figures (Fig 6).

- Line 185: Where is Table 1? I suspect this is a typo and it should say S1? If this is not a typo Table 1 was missing from the reviewer materials. Table 1 is also mentioned on lines 236, 241, and 252.

Sorry, this was also a typo, and should have said S1 Table as opposed to Table 1. This has been updated in the manuscript accordingly. 

- Line 195: space missing between “The” and “1.51%”

Thank you. That has been changed and updated. 

- Figure 3 & 4 (and corresponding supplementary figures): I think adding a colour legend to the maps would help to clarify the figures (instead of identifying different colours in individual maps)

A color legend has been added to Figures 3 and 4 (now Figures 4 and 5). Supplementary Figures 1-8 (now Figures 4-11) have also been changed accordingly.

- Figure 5 (and corresponding supplementary figures): The yellow colour chosen here is hard to see, especially when only a few cells are coloured (such as in East Asia). If the authors could change the colour to something more readable (perhaps a darker orange) it would greatly improve the figures. I think subheading would greatly help here as well (if possible).

Figure 5 (now Figure 6) has been updated to have darker orange colored cells and subheadings. Supplementary Figures 9 and 10 (now Figures 12 and 13) have also been changed accordingly. 

Reviewer #2: Overview:

This paper presents a well-written and very straightforward overlap analysis between different diversity measures for elasmobranchs, i.e. shark, rays and chimeras. It addresses a question that has been asked and answered for other species groups but not yet for elasmobranchs: whether hotspots of species richness also capture evolutionarily distinct, endemic or threatened species reasonably well. The authors find that evolutionary distinctiveness is well captured by total richness, but endemicity less so, and threatened species even less so. This adds to an ongoing discussion about using species richness as a proxy for conservation planning. The authors also identify a few areas where all measures overlap well; although these areas are small in extent they cover almost two thirds (64%) of all elasmobranch species. I think more could be made of this latter finding, which actually surprised me. These results is highly relevant for global conservation planning.

We have added a sentence to the beginning of the discussion to highlight this finding (lines 302-304 in marked up document).

Major concerns:

My two major concerns are as follows:

1. The analysis is entirely focused on coastal species, because most of elasmobranch diversity is coastal. Earlier work has shown, however, that coastal and pelagic species show very different diversity patterns (e.g. Tittensor et al. 2010, Nature). They also likely face different threats and are managed by different entities (i.e. national governments vs. RFMOs). Therefore I think that the analysis should be repeated by separating coastal from pelagic species. The latter will show quite different patterns of richness and different overlap patterns.

We agree that coastal and pelagic species face different threats and are managed by different entities. However, we flag that the greatest diversity of pelagic species can occur along shelf edges and indeed many pelagic species are also neritic – occurring on the shelf. Here, our focus is congruency of different richness measures to aid in spatial planning, mainly in EEZs, not the high seas. Hence, we have chosen to retain pelagic species to capture the true richness and evolutionary distinctiveness occurring in the shelf seas. 

The analysis was also in essence a global study, however, the low richness and wide ranging nature of species means no hotspots were identified in the pelagic ecosystem. Further, endemic species richness tends to be strictly coastal, meaning that separating the analysis into coastal and pelagic would inevitably drop endemicity from the analysis in the pelagic regions, unless endemicity was defined in a different manner. We have added this detail to the methods in lines 134 – 137 and the discussion in lines 422-427 of the marked-up document. We have also included three figures (Fig 1 in the manuscript and Figs S1 and S2 in the SOM) that illustrate the spatial distributional patterns of all of the species richness metrics (total richness, evolutionary distinctiveness, endemicity, and the threatened subsets) at the three varying spatial resolutions (1°, 4°, and 8°). These figures are to help visually clarify the global extent of this study, as well as highlight where the hotspot areas are occurring in comparison to the overall distributional pattern for each of the species richness measures.

2. The analysis is entirely based on IUCN range data which are (as far as I know) largely based on expert opinion. The discussion needs a thorough section on caveats and limitations in this data source. As it, it is treated without any reservations. But it’s unlikely that these maps represent current richness, especially for threatened species which are likely to have undergone significant range extensions. I urge the author to think about how these limitations may have impacted their analysis.

We have added a paragraph on the advantages, limitations, and caveats of this data source to the discussion (lines 393 - 420 in the marked-up document). While we recognize there have been range contractions, our approach to identify the historic pattern of richness for each species and demonstrate a baseline understanding of global shark and ray biodiversity. 

Minor comments:

L41 .. other dimensions of interpretations (not sue what is meant here, revise wording)

Removed “of interpretations”

L80/81; Please update those figures to 2020 using WCMC data base, MPA Atlas or both

These values have been updated to the UNEP-WCMC and IUCN marine protected planet datafiles accessed through www.protectedplanet.net (reference 21).

Introduction is well written and comprehensive, could be tightened up a bit

The introduction has been tightened up where we saw possible. We have removed redundant sentences (i.e. lines 57-58, 63-64, and 71-72) and made other sentences clearer cut by reducing or changing the vocabulary (i.e. lines 54-57, 79-82).

Figure 1 is hard to read and understand, please think about ways to improve clarity.

Figure 1 has been changed and separated into Figure 2 and S3 Fig from suggestions of reviewer #1. These suggestions were to break up Figure 1 so that Figure 1a stands alone and Figure 1b,c are presented as a separate figure in the supplementary materials (as S3 Fig) to help improve clarity. We also added minor tick marks on the y-axis as per reviewer #1’s suggestions.

6. PLOS authors have the option to publish the peer review history of their article (what does this mean?). If published, this will include your full peer review and any attached files.

Do you want your identity to be public for this peer review? For information about this choice, including consent withdrawal, please see our Privacy Policy.

Reviewer #1: No

Reviewer #2: No

Extra Notes:

- We have incorporated a paragraph that discusses the considerations of spatial resolution when performing spatial analyses that was not included in the original submission of this paper. We believe it touches on a crucial aspect of spatial analyses that our results are able to highlight and can help direct future research in determining appropriate resolutions to work with (lines 372 – 391 of marked-up document).

---

## [Editor Report · Decision Letter 1]

12 Jun 2020

PONE-D-20-05961R1

Spatially congruent sites of importance for global shark and ray biodiversity

PLOS ONE

Dear Dr. Derrick,

Thank you for submitting your manuscript to PLOS ONE. After careful consideration, we feel that it has merit but does not fully meet PLOS ONE’s publication criteria as it currently stands. Therefore, we invite you to submit a revised version of the manuscript that addresses the points raised during the review process.

I identified two issues (see below) that must be dealt with before this manuscript can be accepted.

We look forward to receiving your revised manuscript.

Kind regards,

William David Halliday, Ph.D.

Academic Editor

PLOS ONE

Additional Editor Comments (if provided):

Thank you for responding to all reviews in such a detailed manner. I am generally satisfied with your responses, but I identified two issues:

1) In response to Reviewer 1, you stated that because IUCN data are freely available, that you were not required to make your data available. This is not correct, and this is what the reviewer was pointing out. You started with the IUCN data, but then derived multiple data products (i.e. species richness, endemicity, etc) at different spatial resolutions for all species and just for threatened species - essentially all of the data that were used to generate your figures. All of these data products must be made freely available, ideally put in a data repository. Please state explicitly how you will make your data available.

2) Just one minor typo that I caught in the text: line 405, "rarely not vetted" should likely be "rarely vetted".

---

## [Author Response · Author response to Decision Letter 1]

16 Jun 2020

The data used for this analysis were the distribution maps provided by the IUCN. These distributional maps are published on an open access website and freely available for non-commercial use (https://www.iucnredlist.org/terms/terms-of-use), which we have cited appropriately in our manuscript. Since we derived multiple data products at varying spatial resolutions from the IUCN databank, we will upload all relevant data (in the form of shapefiles) that were used for the analysis to the primary author’s GitHub repository under the name of "SpatiallyCongruentSites_2020" once the manuscript has been accepted (https://github.com/daniellederrickh). 

We also edited the typo found on line 405 and changed the text from "rarely not vetted" to "rarely vetted".

---

## [Editor Report · Decision Letter 2]

18 Jun 2020

Spatially congruent sites of importance for global shark and ray biodiversity

PONE-D-20-05961R2

Dear Dr. Derrick,

We’re pleased to inform you that your manuscript has been judged scientifically suitable for publication and will be formally accepted for publication once it meets all outstanding technical requirements.

Kind regards,

William David Halliday, Ph.D.

Academic Editor

PLOS ONE
---

## [Editor Report · Acceptance letter]

23 Jun 2020

PONE-D-20-05961R2 

Spatially congruent sites of importance for global shark and ray biodiversity 

Dear Dr. Derrick:

I'm pleased to inform you that your manuscript has been deemed suitable for publication in PLOS ONE. Congratulations! Your manuscript is now with our production department. 

Kind regards, 

on behalf of

Dr. William David Halliday 

Academic Editor

PLOS ONE